# Locally Private Learning without Interaction Requires Separation

**Amit Daniely**
Hebrew University and Google Research

**Vitaly Feldman***
Google Research

## Abstract

We consider learning under the constraint of local differential privacy (LDP). For many learning problems known efficient algorithms in this model require many rounds of communication between the server and the clients holding the data points. Yet multi-round protocols are prohibitively slow in practice due to network latency and, as a result, currently deployed large-scale systems are limited to a single round. Despite significant research interest, very little is known about which learning problems can be solved by such non-interactive systems. The only lower bound we are aware of is for PAC learning an artificial class of functions with respect to a uniform distribution [39].

We show that the margin complexity of a class of Boolean functions is a lower bound on the complexity of any non-interactive LDP algorithm for distribution-independent PAC learning of the class. In particular, the classes of linear separators and decision lists require exponential number of samples to learn non-interactively even though they can be learned in polynomial time by an interactive LDP algorithm. This gives the first example of a natural problem that is significantly harder to solve without interaction and also resolves an open problem of Kasiviswanathan et al. [39]. We complement this lower bound with a new efficient learning algorithm whose complexity is polynomial in the margin complexity of the class. Our algorithm is non-interactive on labeled samples but still needs interactive access to unlabeled samples. All of our results also apply to the statistical query model and any model in which the number of bits communicated about each data point is constrained.

## 1 Overview

We consider learning in distributed systems where each client $i$ (or user) holds a data point $z_i \in Z$ drawn i.i.d. from some unknown distribution $P$ and the goal of the server is to solve some statistical learning problem using the data stored at the clients. In addition, the communication from the client to the server is constrained. The primary model we consider is that of local differential privacy (LDP) [39]. In this model each user $i$ applies a differentially-private algorithm to their point $z_i$ and then sends the result to the server. The specific algorithm applied by each user is determined by the server. In the general version of the model the server can determine which algorithm the user should apply on the basis of all the previous communications the server has received. In practice, however waiting for the client's response often takes a relatively large amount of time. Therefore in such systems it is necessary to limit the number of rounds of interaction. That is, the queries of the server need to be split into a small number of batches such that the LDP algorithms used in each batch depend only on responses to queries in previous batches (a query specifies the algorithm to apply). Indeed, currently

deployed systems that use local differential privacy use very few rounds (usually just one) [28, 3, 19]. See Section 2 for a formal definition of the model.

In this paper we will focus on the standard PAC learning of a class of Boolean functions $C$ over some domain $X$. In this setting the input distribution $P$ is over labeled examples $(x, y) \in X \times \{-1, 1\}$ where $x$ is drawn from some distribution $D$ and $y = f(x)$ for some unknown $f \in C$ (referred to as the target function). The goal of the learning algorithm is to output a function $h$ such that the error $\mathbf{Pr}_{x \sim D}[f(x) \neq h(x)]$ is small. In the distribution-independent setting $D$ is not known to the learning algorithm while in the distribution-specific setting the learning algorithm only needs to succeed for some specific $D$.

For many of the important classes of functions all known LDP learning algorithms require many rounds of interaction. Yet there are no results that rule out solving these problems without interaction. This problem was first addressed by Kasiviswanathan et al. [39] who demonstrated existence of an artificial class of Boolean functions $C$ over $\{0, 1\}^d$ with the following property. $C$ can be PAC learned efficiently relative to the uniform distribution over $\{0, 1\}^d$ by an interactive LDP protocol but requires $2^{\Omega(d)}$ samples to learn by any non-interactive learning algorithm. The class $C$ is highly unnatural. It splits the domain into two parts. Target function learned on the first half gives the key to the learning problem on the second half of the domain. That problem is exponentially hard to solve without the key. This approach does not extend to distribution-independent learning setting (intuitively, the learning algorithm will not be able to obtain the key if the distribution does not place any probability on the first half of the domain).

Deriving a technique that applies to distribution independent learning is posed as a natural open problem in this area [39]. Even beyond PAC learning, there are no examples of natural problems that provably require exponentially more samples to solve non-interactively.

## 1.1 Our results

We give a new technique for proving lower bounds on the power of non-interactive LDP algorithms for distribution-independent PAC learning. Our technique is based on a connection between the power of interaction and margin complexity of Boolean function classes that we establish. The margin complexity of a class of Boolean functions $C$, denoted by $\mathsf{MC}(C)$, is the inverse of the largest margin of separation achievable by an embedding of $X$ in $\mathbb{R}^d$ that makes the positive and negative examples of each function in $C$ linearly separable (see Definition 2.5). It is a well-studied measure of complexity of classes of functions and corresponding sign matrices in learning theory and communication complexity (*e.g.* [44, 2, 13, 33, 12, 48, 42, 38]).

We prove that only classes that have polynomially small *margin complexity* can be efficiently PAC learned by a non-interactive LDP algorithm. Our lower bound implies that two natural and well-studied classes of functions: linear separators and decision lists require an exponential number of samples to learn non-interactively. Importantly, it is known that these classes can be learned efficiently by interactive LDP algorithms (this follows from the results for the statistical query model that we discuss later). Thus our result gives an exponential separation between the power of interactive and non-interactive protocols. To the best of our knowledge this is the only known such separation for a natural statistical problem (see Section 1.2 for a more detailed comparison with related notions of non-interactive algorithms).

Our result follows from a stronger lower bound that also holds against algorithms for which only the queries that depend on the label of the point are non-interactive (also referred to as *non-adaptive* in related contexts). We will refer to such algorithms as *label-non-adaptive* LDP algorithms. Formally, our lower bounds for such algorithms is as follows. We say that a class of Boolean ($\{-1, 1\}$-valued) functions $C$ is closed under negation if for every $f \in C$, $-f \in C$.

**Theorem 1.1.** *Let $C$ be a class of Boolean functions closed under negation. Assume that there exists a label-non-adaptive $\epsilon$-LDP algorithm $\mathcal{A}$ that, with success probability at least $2/3$, PAC learns $C$ distribution-independently with error less than $1/2$ using at most $n$ examples. Then $n = \Omega(\mathsf{MC}(C)^{2/3}/e^\epsilon)$.*

Our second contribution is an algorithm for learning large-margin linear separators that matches (up to polynomial factors) our lower bound.

**Theorem 1.2.** *Let $C$ be an arbitrary class of Boolean functions over $X$. For any $\alpha, \epsilon > 0$ and $n = \operatorname{poly}\left(\mathsf{MC}(C)/(\alpha\epsilon)\right)$ there is a label-non-adaptive $\epsilon$-LDP algorithm that PAC learns $C$ distribution-independently with accuracy $1 - \alpha$ using at most $n$ examples.*

Learning of large-margin classifiers is a classical learning problem and various algorithms for the problem are widely used in practice. Our learning algorithm is computationally efficient as long as an embedding of $C$ into a $d = \operatorname{poly}\left(\mathsf{MC}(C) \log |X|\right)$-dimensional space can be computed efficiently (such an embedding is known to exists by the Johnson-Lindenstrauss random projection argument [4]). Together these results show an equivalence (up to polynomials) between margin complexity and PAC learning with this limited form of interaction in the LDP model.

Another implication of Theorem 1.2 is that if the distribution over $X$ is fixed (and known to the learning algorithm) then the learning algorithm becomes non-interactive.

**Corollary 1.3.** *Let $C$ be a class of Boolean functions over $X$ and $D$ be an arbitrary distribution over $X$. For any $\alpha, \epsilon > 0$ and $n = \operatorname{poly}\left(\mathsf{MC}(C)/(\alpha\epsilon)\right)$ there is a non-interactive $\epsilon$-LDP algorithm that PAC learns $C$ relative to $D$ with accuracy $1 - \alpha$ using at most $n$ examples.*

**Techniques:** Following the approach of Kasiviswanathan et al. [39], we use the characterization of LDP protocols using the *statistical query* (SQ) model of Kearns [40]. In this model an algorithm has access to a statistical query oracle for $P$ in place of i.i.d. samples from $P$. The most commonly studied SQ oracle give an estimate of the mean of any bounded function with fixed tolerance.

**Definition 1.4.** *Let $P$ be a distribution over a domain $Z$ and $\tau > 0$. A statistical query oracle $STAT_P(\tau)$ is an oracle that given as input any function $\phi \colon Z \to [-1, 1]$, returns some value $v$ such that $|v - \mathbf{E}_{z \sim P}[\phi(z)]| \leq \tau$.*

Tolerance $\tau$ of statistical queries roughly corresponds to the number of random samples in the traditional setting. Non-adaptive (or non-interactive) SQ algorithms are defined analogously to LDP protocols. The reductions between learning in the SQ model and learning in the LDP model given by Kasiviswanathan et al. [39] preserve the number of rounds of interaction of a learning algorithm.

The key technical tool we apply to prove our lower bound is a result of Feldman [30] relating margin complexity and a certain notion of complexity for statistical queries. The result shows that the existence of a (possibly randomized) algorithm that outputs a set $T$ of $m$ functions such that for every $f \in C$ and distribution $D$, with significant probability one of the functions in $T$ is at least $1/m$-correlated with $f$ relative to $D$ implies that $\mathsf{MC}(C) = O(m^{3/2})$ (the sharpest bound was proved in [38]). We then show that such a set of functions can be easily extracted from the queries of any label-non-adaptive SQ algorithm for learning $C$.

Our label-non-adaptive LDP learning algorithm for large-margin halfspaces relies on a new formulation of halfspace learning as a stochastic convex optimization problem. The crucial property of this program is that (approximately) computing sub-gradients can be done by using a fixed set of non-adaptive queries (that measure the correlation of each of the attributes with the label) and (adaptive) but label-independent queries. We can then use an arbitrary gradient-descent-based LDP algorithm for stochastic convex optimization. Such algorithms were first described by Duchi et al. [21]. For simplicity, we appeal to the fact that such algorithms can also be implemented in the statistical query model [32].

**Corollaries:** The class of decision lists (see [47, 41] for a definition) and the class of linear separators (or halfspaces) over $\{0, 1\}^d$ are known to have exponentially large margin complexity [34, 15, 48] (and are also negation closed). In contrast, these classes are known to be learnable efficiently by SQ algorithms [40, 24] and thus also by LDP algorithms. Formally, we obtain the following lower bounds:

**Corollary 1.5.** *Any label-non-adaptive $\epsilon$-LPD algorithm that PAC learns the class of linear separators over $\{0, 1\}^d$ with error less than $1/2$ and success probability at least $3/4$ must use $n = 2^{\Omega(d)}/e^\epsilon$ i.i.d. examples. For learning the class of decision lists under the same conditions the algorithm must use $n = 2^{\Omega(d^{1/3})}/e^\epsilon$ i.i.d. examples.*

Our use of the statistical query model to prove the results implies that we can derive the analogues of our results in other models that have connections to the SQ model. One of such models is the distributed model in which only a small number of bits is communicated from each client. Namely,

each client applies a function with range $\{0, 1\}^k$ to their input and sends the result to the server (for some $k \ll \log |Z|$). As in the case of LDP, the specific function used is chosen by the server. One motivation for this model is collection of data from remote sensors where the cost of communication is highly asymmetric. In the context of learning this model was introduced by Ben-David and Dichterman [11] and generalized by Steinhardt et al. [50]. Identical and closely related models are often studied in the context of distributed statistical estimation with communication constraints (*e.g.* [43, 45, 46, 57, 51, 53, 1]). As in the setting of LDP, the number of rounds of interaction that the server uses to solve a learning problem in this model is a critical resource. Using the equivalence between this model and SQ learning that preserves the number of rounds of interaction we immediately obtain analogous results for this model. We are not aware of any prior results on the power of interaction in the context of this model. See Section 5 for additional details.

## 1.2 Related work

Smith et al. [49] address the question of the power of non-interactive LDP algorithms in the closely related setting of stochastic convex optimization. They derive new non-interactive LDP algorithms for the problem albeit requiring an exponential in the dimension number of queries. They also give an exponential lower bound for non-interactive algorithms that are further restricted to obtain only local information about the optimized function. Subsequently, upper and lower bounds on the number of queries to the gradient/second-order oracles for algorithms with few rounds of interaction have been studied by several groups [23, 56, 7, 18]. In the context of discrete optimization from queries for the value of the optimized function the round complexity has been recently investigated in [8, 6, 9]. To the best of our knowledge, the techniques used in these works are unrelated to ours. Also in all these works the lower bounds rely heavily on the fact that the oracle provides only local (in the geometric sense) information about the optimized function. In contrast, statistical queries allow getting global information about the optimized function.

A number of lower bounds on the sample complexity of LDP algorithms demonstrate that LDP is less efficient than the central model of differential privacy (*e.g.* [22, 20]). The number of data samples necessary to answer statistical queries chosen adaptively has recently been studied in a line of work on adaptive data analysis [27, 35, 10, 52]. Our work provably demonstrates that the use of such adaptive queries is important for solving basic learning problems.

**Subsequent work:** Acharya et al. [1] implicitly give a separation between interactive and non-interactive protocols for the problem of identity testing for a discrete distribution over $k$ elements, albeit a relatively weak one ($O(k)$ vs $\Omega(k^{3/2})$ samples). The work of Joseph et al. [36, 37] explores a different aspect of interactivity in LDP. Specifically, they distinguish between two types of interactive protocols: fully-interactive and sequentially-interactive ones. Fully-interactive protocols place no restrictions on interaction whereas sequentially-interactive ones only allows asking one query per user. They give a separation showing that sequentially-interactive protocols may require exponentially more samples than fully interactive ones. This separation is orthogonal to ours since our lower bounds are against completely non-interactive protocols and we separate them from sequentially-interactive protocols.

## 2 Preliminaries

For integer $n \geq 1$ let $[n] \doteq \{1, \ldots, n\}$.

**Local differential privacy:** In the local differential privacy (LDP) model [55, 29, 39] it is assumed that each data sample obtained by the server is randomized in a differentially private way. This is modeled by assuming that the server running the learning algorithm accesses the dataset via an oracle defined below.

**Definition 2.1** ([39]). *An $\epsilon$-local randomizer $R : Z \to W$ is a randomized algorithm that satisfies $\forall z_1, z_2 \in Z$ and $w \in W$, $\mathbf{Pr}[R(z_1) = w] \leq e^\epsilon \mathbf{Pr}[R(z_2) = w]$. For a dataset $S \in Z^n$, an $\mathrm{LR}_S$ oracle takes as an input an index $i$ and a local randomizer $R$ and outputs a random value $w$ obtained by applying $R(z_i)$. An algorithm is (compositionally) $\epsilon$-LDP if it accesses $S$ only via the $\mathrm{LR}_S$ oracle with the following restriction: for all $i \in [n]$, if $\mathrm{LR}_S(i, R_1), \ldots, \mathrm{LR}_S(i, R_k)$ are the algorithm's invocations of $\mathrm{LR}_S$ on index $i$ where each $R_j$ is an $\epsilon_j$-randomizer then $\sum_{j \in [k]} \epsilon_j \leq \epsilon$.*

For a non-interactive LDP algorithm one can assume without loss of generality that each sample is queried only once since the application of $k$ fixed local randomizers can be equivalently seen as an execution of a single $\epsilon$-randomizer with $\sum_{j \in [k]} \epsilon_j \leq \epsilon$. Further, in this definition the privacy parameter is defined as the composition of the privacy parameters of all the randomizers. A more general (and less strict) way to define the privacy parameter of an LDP protocol is as the differential privacy of the entire transcript of the protocol (see [36] for a more detailed discussion). This distinction does not affect our results since in our lower and upper bounds each sample is only queried once. For such protocols these two ways to measure privacy coincide. The local model of privacy can be contrasted with the standard, or central, model of differential privacy where the entire dataset is held by the learning algorithm whose output needs to satisfy differential privacy [25]. This is a stronger model and an $\epsilon$-LPD algorithm also satisfies $\epsilon$-differential privacy.

**Equivalence to statistical queries:** The statistical query model of Kearns [40] is defined by having access to $\text{STAT}_P(\tau)$ oracle, where $P$ is the unknown data distribution. To solve a learning problem in this model an algorithm needs to succeed for any valid (that is satisfying the guarantees on the tolerance) oracle's responses. In other words, the guarantees of the algorithm should hold in the worst case over the responses of the oracle. A randomized learning algorithm needs to succeed for any SQ oracle whose responses may depend on the all queries asked so far but not on the internal randomness of the learning algorithm.

A special case of statistical queries are counting or linear queries in which the distribution $P$ is uniform over the elements of a given database $S \in Z^n$. In other words the goal is to estimate the empirical mean of $\phi$ on the given set of data points. This setting is studied extensively in the literature on differential privacy (see [26] for an overview) and our discussion applies to this setting as well.

For an algorithm in LDP and SQ models we say that the algorithm is *non-interactive* (or *non-adaptive*) if all its queries are determined before observing any of the oracle's responses. Similarly, we say that the algorithm is *label-non-adaptive* if all the queries that depend on oracle's response are label-independent (the query function depends only on the point).

Kasiviswanathan et al. [39] show that one can simulate $\text{STAT}_P(\tau)$ oracle with success probability $1-\delta$ by an $\epsilon$-LDP algorithm using $\text{LR}_S$ oracle for $S$ containing $n = O(\log(1/\delta)/(\epsilon\tau)^2)$ i.i.d. samples from $P$. This has the following implication for simulating SQ algorithms.

**Theorem 2.2** ([39]). *Let $\mathcal{A}_{SQ}$ be an algorithm that makes at most $t$ queries to $\text{STAT}_P(\tau)$. Then for every $\epsilon > 0$ and $\delta > 0$ there is an $\epsilon$-local algorithm $\mathcal{A}$ that uses $\text{LR}_S$ oracle for $S$ containing $n \geq n_0 = O(t\log(t/\delta)/(\epsilon\tau)^2)$ i.i.d. samples from $P$ and produces the same output as $\mathcal{A}_{SQ}$ (for some valid answers of $\text{STAT}_P(\tau)$) with probability at least $1 - \delta$. Further, if $\mathcal{A}_{SQ}$ is non-interactive then $\mathcal{A}$ is non-interactive.*

Kasiviswanathan et al. [39] also prove a converse of this theorem.

**Theorem 2.3** ([39]). *Let $\mathcal{A}$ be an $\epsilon$-LPD algorithm that makes at most $t$ queries to $\text{LR}_S$ for $S$ drawn i.i.d. from $P^n$. Then for every $\delta > 0$ there is an SQ algorithm $\mathcal{A}_{SQ}$ that in expectation makes $O(t \cdot e^\epsilon)$ queries to $\text{STAT}_P(\tau)$ for $\tau = \Theta(\delta/(e^{2\epsilon}t))$ and produces the same output as $\mathcal{A}$ with probability at least $1 - \delta$. Further, if $\mathcal{A}$ is non-interactive then $\mathcal{A}_{SQ}$ is non-interactive.*

**PAC learning and margin complexity:** Our results are for the standard PAC model of learning [54].

**Definition 2.4.** *Let $X$ be a domain and $C$ be a class of Boolean functions over $X$. An algorithm $\mathcal{A}$ is said to PAC learn $C$ with error $\alpha$ if for every distribution $D$ over $X$ and $f \in C$, given access (via oracle or samples) to the input distribution over examples $(x, f(x))$ for $x \sim D$, the algorithm outputs a function $h$ such that $\mathbf{Pr}_D[f(x) \neq h(x)] \leq \alpha$ with probability at least $2/3$.*

We say that the learning algorithm is efficient if its running time is polynomial in $\log|X|$, $\log|C|$ and $1/\epsilon$.

For dimension $d$, we denote by $\mathcal{B}^d(1)$ the unit ball in $\ell_2$ norm in $\mathbb{R}^d$.

**Definition 2.5.** *Let $X$ be a domain and $C$ be a class of Boolean functions over $X$. The margin complexity of $C$, denoted $\text{MC}(C)$, is the minimal number $M \geq 0$ such that for some $d$, there is an embedding $\Psi : X \to \mathcal{B}^d(1)$ for which the following holds: for every $f \in C$ there is $w \in \mathcal{B}^d(1)$ such*

*that*

$$\min_{x \in X}\{f(x) \cdot \langle w, \Psi(x) \rangle\} \geq \frac{1}{M}.$$

As pointed out in [30], margin complexity[2] is equivalent (up to a polynomial) to the existence of a (possibly randomized) algorithm that outputs a small set of functions such that with significant probability one of those functions is correlated with the target function. The upper bound in [30] was sharpened by Kallweit and Simon [38] although they proved it only for deterministic algorithms (which corresponds to a single fixed set of functions and is referred to as the CSQ dimension). It is however easy to see that their sharper bound extends to randomized algorithms with an appropriate adjustment of the bound and we give the resulting statement below:

**Lemma 2.6** ([30, 38])**.** *Let $X$ be a domain and $C$ be a class of Boolean functions over $X$. Assume that there exists a (possibly randomized) algorithm $\mathcal{A}$ that generates a set of functions $h_1, \ldots, h_m$ satisfying: for every $f \in C$ and distribution $D$ over $X$ with probability at least $\beta > 0$ (over the randomness of $\mathcal{A}$) there exists $i \in [m]$ such that $|\mathbf{E}_{x \sim D}[f(x)h_i(x)]| \geq 1/m$. Then*

$$\mathsf{MC}(C) \leq \frac{2}{\beta}m^{3/2}.$$

The conditions in Lemma 2.6 are also known to be necessary for low margin complexity.

**Lemma 2.7** ([30, 38])**.** *Let $X$ be a domain, $C$ be a class of Boolean functions over $X$ and $d = \mathsf{MC}(C)$. Then for $m = O(\ln(|C||X|)d^2)$, there exists a set of functions $h_1, \ldots, h_m$ satisfying: for every $f \in C$ and distribution $D$ over $X$ there exists $i \in [m]$ such that $|\mathbf{E}_{x \sim D}[f(x)h_i(x)]| \geq 1/m$.*

## 3 Lower bounds for label-non-adaptive algorithms

We prove the SQ version of our lower bound. Theorem 1.1 then follows immediately by applying the simulation result from Theorem 2.3.

**Theorem 3.1.** *Let $C$ be a class of Boolean functions closed under negation. Assume that for some $m$ there exists a label-non-adaptive possibly randomized SQ algorithm $\mathcal{A}$ that, with success probability at least $2/3$, PAC learns $C$ distribution-independently with error less than $1/2$ using at most $m$ queries to STAT$(1/m)$. Then $\mathsf{MC}(C) \leq 6m^{3/2}$.*

*Proof.* We first recall a simple observation from [14] that allows to decompose each statistical query into a correlational and label-independent parts. Namely, for a function $\phi \colon X \times \{-1, 1\} \to [-1, 1]$,

$$\phi(x, y) = \frac{1-y}{2}\phi(x, -1) + \frac{1+y}{2}\phi(x, 1) = \frac{\phi(x, -1) + \phi(x, 1)}{2} + y \cdot \frac{\phi(x, 1) - \phi(x, -1)}{2}.$$

For a query $\phi$, we will use $h$ and $g$ to denote the parts of the decomposition $\phi(x, y) = g(x) + yh(x)$:

$$h(x) \doteq \frac{\phi(x, 1) - \phi(x, -1)}{2}$$

and

$$g(x) \doteq \frac{\phi(x, 1) + \phi(x, -1)}{2}.$$

For every input distribution $D$ and target functions $f$, we define the following SQ oracle. Given a query $\phi$, if $|\mathbf{E}_D[f(x)h(x)]| \geq 1/m$ then the oracle provides the exact expectation $\mathbf{E}_D[\phi(x, f(x)]$ as the response. Otherwise, it answers with $\mathbf{E}_D[g(x)]$. Note that, by the properties of the decomposition, this is a valid implementation of the SQ oracle.

Let $\mathcal{A}(r)$ denote $\mathcal{A}$ with its random bits set to $r$, where $r$ is drawn from some distribution $R$. Let $\phi_1^r, \ldots, \phi_{m'}^r \colon X \times \{-1, 1\} \to [-1, 1]$ be the statistical queries asked by $\mathcal{A}(r)$ that depend on the label (where $m' \leq m$). Note that, by the definition of a label-non-adaptive SQ algorithm, all these

queries are fixed in advance and do not depend on the oracle's answers. Let $g_i^r$ and $h_i^r$ denote the decomposition of these queries into correlational and label-independent parts. Let $h_{f,D}^r$ denote the hypothesis output by $\mathcal{A}(r)$ when used with the SQ oracle defined above.

We claim that if $\mathcal{A}$ achieves error $< 1/2$ with probability at least $2/3$, then for every $f \in C$ and distribution $D$, with probability at least $1/3$, there exists $i \in [m']$ such that $|\mathbf{E}_D[f(x)h_i^r(x)]| \geq 1/m$ (satisfying the conditions of Lemma 2.6 with $\beta = 1/3$). To see this, assume for the sake of contradiction that for some distribution $D$ and function $f \in C$,

$$\Pr_{r \sim R}[r \in T(f,D)] > 2/3,$$

where $T(f,D)$ is the set of all random strings $r$ such that for all $i \in [m']$, $|\mathbf{E}_D[f(x)h_i^r(x)]| < 1/m$. Let $S(f,D)$ denote the set of random strings $r$ for which $\mathcal{A}$ succeeds (with the given SQ oracle), that is $\mathbf{Pr}_D[f(x) \neq h_{f,D}^r(x)] < 1/2$.

By our assumption, $\mathbf{Pr}_{r \sim R}[r \in S(f,D)] \geq 2/3$ and therefore

$$\Pr_{r \sim R}[r \in T(f,D) \cap S(f,D)] > 1/3. \tag{1}$$

Now, observe that $T(-f,D) = T(f,D)$ and, in particular, the answers of our SQ oracle to $\mathcal{A}(r)$'s queries are identical for $f$ and $-f$ whenever $r \in T(f,D)$. Further, if $\mathbf{Pr}_D[f(x) \neq h_{f,D}^r(x)] < 1/2$ then $\mathbf{Pr}_D[-f(x) \neq h_{f,D}^r(x)] > 1/2$. This means that for every $r \in T(f,D) \cap S(f,D)$, $\mathcal{A}(r)$ fails for the target function is $-f$ and the distribution $D$ (by definition, $-f \in C$). By eq. (1) we obtain that $\mathcal{A}$ fails with probability $> 1/3$ for $-f$ and $D$. This contradicts our assumption and therefore we obtain that

$$\Pr_{r \sim R}[r \notin T(f,D)] \geq 1/3.$$

By Lemma 2.6, we obtain the claim. □

## 3.1 Applications

We will now spell out several easy corollaries of our lower bound, simulation results and existing SQ algorithms. Together they imply the claimed separations for halfspaces and decision lists. We start with the class of halfspaces over $\{0,1\}^d$ which we denote by $C_{HS}$. The lower bound on the margin complexity of halfspaces is implied by a celebrated work of Goldmann et al. [34] on the complexity of linear threshold circuits (the connection of this result to margin complexity is due to Sherstov [48]):

**Theorem 3.2** ([34, 48])**.** $\mathsf{MC}(C_{HS}) = 2^{\Omega(d)}$.

We denote the class of decision lists over $\{0,1\}^d$ by $C_{DL}$ (see [41] for a standard definition). A lower bound on the margin complexity decision lists was derived by Buhrman et al. [15] in the context of communication complexity.

**Theorem 3.3** ([15])**.** $\mathsf{MC}(C_{DL}) = 2^{\Omega(d^{1/3})}$.

Combining these results with Theorem 1.1 we obtain the lower bound on complexity of LDP algorithms for learning linear classifiers and decision lists given in Corollary 1.5.

Learnability of decision list using statistical queries is a classical result of Kearns [40]. Applying the simulation in Theorem 2.2 we obtain polynomial time learnability of this class by (interactive) LDP algorithms.

**Theorem 3.4** ([40])**.** *For every $\epsilon, \alpha > 0$, there exists an $\epsilon$-LDP learning algorithm that PAC learns $C_{DL}$ with error $\alpha$ using $\mathrm{poly}(d/(\epsilon\alpha))$ i.i.d. examples (with one query per example).*

In the case of halfspaces, Dunagan and Vempala [24] give the first efficient algorithm for PAC learning halfspaces (their description is not in the SQ model but it is known that their algorithm can be easily converted to the SQ model [5]). Applying Theorem 2.2 we obtain learnability of this class by (interactive) LDP algorithms.

**Theorem 3.5** ([24, 5])**.** *For every $\epsilon, \alpha > 0$, there exists an $\epsilon$-LDP learning algorithm that PAC learns $C_{HS}$ with error $\alpha$ using $\mathrm{poly}(d/(\epsilon\alpha))$ i.i.d. examples (with one query per example).*

# 4 Label-non-adaptive learning algorithm for halfspaces

Our algorithm for learning large-margin halfspaces relies on the formulation of the problem of learning a halfspace as the following convex optimization problem. Proofs of results in this section can be found in the full version [16].

**Lemma 4.1.** *Let $P$ be a distribution on $\mathcal{B}^d(1) \times \{-1, 1\}$. Suppose that there is a vector $w^* \in \mathcal{B}^d(1)$ such that $\mathbf{Pr}_{(x,\ell) \sim P}[\langle w^*, \ell x \rangle \geq \gamma] = 1$. Let $(e_1, \ldots, e_d)$ denote the standard basis of $\mathbb{R}^d$ and let $w$ be a unit vector such that for $\alpha, \beta \in (0, 1)$*

$$F(w) \doteq \mathop{\mathbf{E}}_{(x,\ell) \sim P} \left[ \sum_{i=1}^{d} |\langle w + \gamma e_i, x \rangle| - \langle w + \gamma e_i, \ell x \rangle + \sum_{i=1}^{d} |\langle w - \gamma e_i, x \rangle| - \langle w - \gamma e_i, \ell x \rangle \right] \leq \alpha\beta. \tag{2}$$

*Then, $F(w^*) = 0$ and*

$$\mathop{\mathbf{Pr}}_{P} \left[ \langle w, \ell x \rangle \geq -\frac{\beta}{2} + \frac{\gamma^2}{\sqrt{d}} \right] \geq 1 - \alpha.$$

*In particular, if $\beta < \frac{2\gamma^2}{\sqrt{d}}$ then $\mathbf{Pr}_P[\langle w, \ell x \rangle > 0] \geq 1 - \alpha$.*

We now describe how to solve the convex optimization problem given in Lemma 4.1. Both the running time and accuracy of queries of our solution depend on the ambient dimension $d$. This dimension is not necessarily upper-bounded by a polynomial in $\mathsf{MC}(C)$. However, the well-known random projection argument shows that the dimension can be reduced to $O(\log(1/\delta)/\mathsf{MC}(C)^2)$ at the expense of small multiplicative decrease in the margin and probability of at most $\delta$ of failure (for every individual point over the randomness of the random projection) [4, 12]. This fact together with Markov's inequality implies the following standard lemma:

**Lemma 4.2.** *Let $d$ be an arbitrary dimension. For every $\delta$ and $\gamma$, there exists a distribution $\Psi$ over mappings $\psi \colon \mathcal{B}^d(1) \to \mathcal{B}^{d'}(1)$, where $d' = O(\log(1/\delta)/\gamma^2)$ such that: For every distribution $D$ and function $f$ over $\mathcal{B}^d(1)$, if there exists $w \in \mathcal{B}^d(1)$ such that $\mathbf{Pr}_{x \sim D}[f(x) \cdot \langle w, x \rangle \geq \gamma] = 1$ then*

$$\mathop{\mathbf{Pr}}_{\psi \sim \Psi} \left[ \exists w', \ \mathop{\mathbf{Pr}}_{x \sim D} \left[ f(x) \cdot \langle w', \psi(x) \rangle \geq \frac{\gamma}{2} \right] \geq 1 - \delta \right] \geq 1 - \delta.$$

Lemma 4.2 ensures that at most a tiny fraction $\delta$ of the points (according to $D$) does not satisfy the margin condition. This is not an issue as we will be implementing our algorithm in the SQ model that, by definition allows any of the answers to its queries to be imprecise. The lemma also allows for tiny probability that the mapping will fail altogether (making the overall algorithm randomized).

Therefore the only ingredient missing for establishing Theorem 1.2 is a label-non-adaptive SQ algorithm for solving the convex optimization algorithm in dimension $d$ using a polynomial in $d$, $1/\gamma$ and $1/\alpha$ number of queries and (the inverse of) tolerance:

**Lemma 4.3.** *Let $P$ be a distribution on $\mathcal{B}^d(1) \times \{-1, 1\}$. Suppose that there is a vector $w^* \in \mathcal{B}^d(1)$ such that $\mathbf{Pr}_{(x,\ell) \sim P}[\langle w^*, \ell x \rangle \geq \gamma] = 1$. There is a label-non-adaptive SQ algorithm that for every $\alpha \in (0, 1)$ uses $O(d^4/(\gamma^4 \alpha^2))$ queries to $\mathrm{STAT}_P(\Omega(\gamma^4 \alpha^2/d^3))$, and finds a vector $w$ such that $\mathbf{Pr}_P[\langle w, \ell x \rangle > 0] \geq 1 - \alpha$.*

# 5 Implications for distributed learning with communication constraints

In this section we briefly define the model of bounded communication per sample, state the known equivalence results to the SQ model and spell out the immediate corollary of our lower bound.

In the bounded communication model [11, 50] it is assumed that the total number of bits learned by the server about each data sample is bounded by $\ell$ for some $\ell \ll \log |Z|$. As in the case of LDP this is modeled by using an appropriate oracle for accessing the dataset.

**Definition 5.1.** *We say that an algorithm $R \colon Z \to \{0, 1\}^\ell$ extracts $\ell$ bits. For a dataset $S \in Z^n$, an $\mathrm{COMM}_S$ oracle takes as an input an index $i$ and an algorithm $R$ and outputs a random value $w$ obtained by applying $R(z_i)$. An algorithm is $\ell$-bit communication bounded if it accesses $S$ only via the $\mathrm{COMM}_S$ oracle with the following restriction: for all $i \in [n]$, if $\mathrm{COMM}_S(i, R_1), \ldots, \mathrm{COMM}_S(i, R_k)$ are the algorithm's invocations of $\mathrm{COMM}_S$ on index $i$ where each $R_j$ extracts $\ell_j$ bits then $\sum_{j \in [k]} \ell_j \leq \ell$.*

We use (non-)adaptive in the same sense as we do for LDP.

As first observed by Ben-David and Dichterman [11], it is easy to simulate a single query to $\mathrm{STAT}_P(\tau)$ by extracting a single bit from each of the $O(1/\tau^2)$ samples. This gives the following simulation.

**Theorem 5.2** ([11]). *Let $\mathcal{A}_{SQ}$ be an algorithm that makes at most $t$ queries to $\mathrm{STAT}_P(\tau)$. Then for every $\delta > 0$ there is an $\epsilon$-local algorithm $\mathcal{A}$ that uses $\mathrm{COMM}_S$ oracle for $S$ containing $n \geq n_0 = O(t \log(t/\delta)/\tau^2)$ i.i.d. samples from $P$ and produces the same output as $\mathcal{A}_{SQ}$ (for some valid answers of $\mathrm{STAT}_P(\tau)$) with probability at least $1 - \delta$. Further, if $\mathcal{A}_{SQ}$ is non-interactive then $\mathcal{A}$ is non-interactive.*

The converse of this theorem for the simpler COMM oracle that accesses each sample once was given in [11, 31]. For the stronger oracle in Definition 5.1, the converse was given by Steinhardt et al. [50].

**Theorem 5.3** ([50]). *Let $\mathcal{A}$ be an $\ell$-bit communication bounded algorithm that makes queries to $\mathrm{COMM}_S$ for $S$ drawn i.i.d. from $P^n$. Then for every $\delta > 0$, there is an SQ algorithm $\mathcal{A}_{SQ}$ that makes $2n\ell$ queries to $\mathrm{STAT}_P\left(\delta/(2^{\ell+1}n)\right)$ and produces the same output as $\mathcal{A}$ with probability at least $1 - \delta$. Further, if $\mathcal{A}$ is non-interactive then $\mathcal{A}_{SQ}$ is non-interactive.*

Note that in this simulation we do not need to assume a separate bound on the number of queries since at most $\ell n$ queries can be asked.

A direct corollary of Theorems 3.1 and 5.3 and is the following lower bound:

**Corollary 5.4.** *Let $C$ be a class of Boolean functions closed under negation. Any label-non-adaptive $\ell$-communication bounded algorithm that PAC learns $C$ with error less than $1/2$ and success probability at least $3/4$ using queries to $\mathrm{COMM}_S$ for $S$ drawn i.i.d. from $P^n$ must have $n = \mathsf{MC}(C)^{2/3}/2^\ell$.*

Our other results can be extended analogously.

# 6 Discussion

Our work shows that polynomial margin complexity is a necessary and sufficient condition for efficient distribution-independent PAC learning of a class of binary classifiers by a label-non-adaptive SQ/LDP/limited-communication algorithm. A natural open problem that is left open is whether there exists an efficient and fully non-interactive algorithm for any class of polynomial margin complexity. We conjecture that the answer is "no" and in this case the question is how to characterize the problems that are learnable by non-interactive algorithms. See [17] for a more detailed discussion of this open problem.

A significant limitation of our result is that it does not rule out even a 2-round algorithm for learning halfspaces (or decision lists). This is, again, in contrast to the fact that learning algorithms for these classes require at least $d$ rounds of interaction. We believe that extending our lower bounds to multiple-round algorithms and quantifying the tradeoff between the number of rounds and the complexity of learning is an important direction for future work.

## Footnotes

*Part of this work was done while the author was visiting the Simons Institute for the Theory of Computing.

[1]Extended abstract. Full version appears as [16].

[2]The results there are stated in terms of another notion that is closely related to margin complexity. Namely, the smallest dimension $d$ for which for which there exists a mapping of $X$ to $\{0, 1\}^d$ such that every $f \in C$ becomes expressible as a majority function over some subset $T \subseteq [d]$ of variables. See the discussion in Sec. 6 of [38].

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
