[Reviews · NeurIPS 2019]

Reviewer 1



Summary of Contributions: UPDATE: I have read the author rebuttal and my review is unchanged. This work considers the role of interaction in (distribution-free) PAC learning with local differential privacy (LDP). In this model the learning algorithm only has access to examples that have been locally randomized in a differentially private way; the learner may specify the index of the example and the randomizer. If the learning algorithm makes all of its queries in advance, it is non-interactive, and if it makes its queries independent of the labels returned by the oracle, it is label-non-adaptive. The main results of this paper show that PAC learning with a label-non-adaptive LDP algorithm is characterized (up to polynomial factors) by the margin complexity MC(C) of the class, which is the inverse largest margin of separation achievable by linearly separable (into positive and negative examples) embedding of the domain. The lower bound (see 1. below) has interesting consequences regarding the power of interaction for LDP PAC learning. For example, this implies that any non-interactive (and even label-non-adaptive) algorithm for decision lists requires ~2^Omega(d^⅔) examples; any such algorithm for linear separators requires 2^Omega(d) examples. On the other hand, these classes do have efficient LDP PAC learning algorithms. The last exponential separation result for PAC learning is due to Kasiviswanathan et al. (2011), who first introduced the notion of LDP; they gave an exponential separation for PAC learning a less natural class that does not hold in the distribution-free setting. Thus, this work gives the first distribution-free separation and does so for natural and well-studied learning problems. The characterization via margin complexity is given by the following statements : Any label-non-adaptive eps-LDP algorithm that PAC learns a class C (closed under negations) requires Omega(MC(C)^{⅔} / e^{eps}) examples. There is a label-non-adaptive eps-LDP algorithm that PAC learns a class C to accuracy 1-alpha with poly(MC(C)/alpha*eps) samples. In fact the characterization is for the non-interactive statistical query (SQ) model, which then translates to LDP PAC learning through the equivalence proven in Kasiviswanathan et al. The proof of statement 1 uses a connection between margin complexity and correlational statistical query (CSQ) dimension established by Feldman 2008, which was improved by Kallweit and Simon 2011. These works show that the ability to construct a small set of universally correlated functions for a class C implies small margin complexity. This work shows how to construct such a set using a label-non-adaptive LDP algorithm. The proof of statement 2 is more involved and reformulates the halfspace problem as a stochastic convex optimization one, applies a random projection to the embedding space (which may be large compared to the margin), and shows how to solve the convex optimization with label-non-adaptive queries. Strengths/Weaknesses The power of interaction for solving problems with local differential privacy is an important topic within differential privacy that is motivated by both theoretical and practical concerns. This work gives a strong lower bound for learning natural and well-studied classes without interaction; these classes have efficient algorithms with interaction. On the other hand, Kasiviswanathan gave a strong lower bound for a more contrived class, which only has an efficient algorithm with interaction under the uniform distribution. Since that work does not give a separation for general PAC learning, I view this result as a significant contribution to our understanding of the power of interaction for local differentially private algorithms. As a technical contribution, this work makes a simple but nice connection to the notion of margin complexity (admittedly, a good portion of the connection is already made in Feldman 2008), and shows that margin complexity characterizes non-interactive LDP PAC learning for a more permissive notion of non-interaction. One drawback of this characterization, as noted in the conclusion, is the need for this more permissive definition; another area left open is to show lower bounds for algorithms with bounded interaction. In addition, the paper is of high quality and is generally well-written. A few specific comments regarding clarity follow below. Notes: Lines 72-73: Smith et al. 2017 do give a natural statistical problem which can be solved with interaction and show an exponential (to my knowledge) lower bound without interaction for a restricted but natural class of algorithms (which include the interactive ones). In light of this, it seems fair to point to the exposition in 1.2 with a footnote here. Lines 153: If this bound is exponential say this instead of “strong”? Line 110: It seems one should also cite Kallweit and Simon since that stronger bound, which is proven in a different way than Feldman 2008, is used in the results that have already been stated. Line 152: typo “in the dimension number of queries” Lemma 2.6: In both sources the corresponding Lemma is stated in terms of the CSQ-dimension, which makes it hard to track down for the unfamiliar reader. Furthermore, Feldman 2008 states result without any reference to margin complexity. So, it would be helpful to mention this to the reader or state the location (e.g., see proof of Theorem 5.4 in Feldman 2008). Line 258: label-non-adaptive?

Reviewer 2



Update: authors "We are puzzled by this statement since none of the mentioned works provides any tools or partial progress toward the 7 solution of the problem we consider. Most notably, [JMNR19] considers a separation of completely unrelated nature 8 and is also subsequent to our work." --- I was referring to the distribution specific separation that you reference in the paper. my review is unchanged. originality: the separation between interactive and sequentially interactive is novel, and uses a new technique for lower bounds that relies on margin complexity. There has been similar work on interactivity, and smith does obtain a separation albeit while requiring the optimization algorithm to access only local information. quality: the technical work is sophisticated and at a high level. clarity: the writing is clear, the positioning relative to related work is comprehensive. see detailed comments for specific suggestions on improving the draft. significance: A definite contribution to our understanding of the power of interaction under local differential privacy. Certainly not a breakthrough result however, as separations with slightly stronger assumptions (distribution-specific learning, accessing only local information, separation between SI and Fully Interactive) already existed. **Comments** * Definition 2.1 is actually incorrect. This would be what is defined in [JNMR] as a compositional epsilon-LDP algorithm. It is possible for the transcript to be epsilon dp and the epsilon_j to sum up to more than epsilon. I don't think changing this definition would impact the rest of the paper. * The authors should update the related work and some of the statements to reflect the recent work [https://arxiv.org/abs/1907.00813]. Obviously this was uploaded subsequent to the current paper...they do show an exponential separation between sequential interactivity and full interactivity, which implies an exponential separation between full interactivity and non-interactivity. Note that it does not imply the exponential separation proven here between SI and non-interactive. * The way thm. 1.2 is stated its a bit unclear if its for any class C or a specific class of large-margin linear separators. *I wasn't familiar with margin complexity before... so based on 2.5 if the dual class of f contains the all zeros classifier, e.g. x such that f(x) = 0 for all X, the margin complexity is infinite? *In line 254 shouldn't this be E[f(x)h(x)] > 1/m?

Reviewer 3



This theoretical paper addresses the problem of non-interactive (or non-adaptive) learning under the constraint of Local Differential Privacy (LDP) or equivalently under the setting of Statistical Queries (SQ). It solves an open problem formulated in (Kasiviswanathan et al. 2011 "What can we learn privately ?"). UPDATE: Being non-familiar with the SQ formalism I had some difficulties entering the paper, and my first review was a bit too rough. The contribution is clearly of interest. But I maintain, that the paper is not really self-contained: it is required to read (Kasiviswanathan et al. 2011) to understand the work and the scientific issues it addresses. The structure of the short version lacks a bit of balance between a 4-pages long "Overview" section which stays somehow in-between an informal and a formal statement of the problems, and the rest of the paper where key notions are formalized. The "Preliminary" section starts only at the middle of the paper and uses reference to notions that where defined in the overview. For instance, the notion of epsilon-LDP is formalized only there in Definition 2.1, but is intensively used in the overview. My feeling on that paper is that : pro: - It solves an open problem stated in (Kasiviswanathan et al. 2011) and relies strongly on this paper 's results; - The maths that I checked in section 3 seems solid. con: - The formalism is a bit heavy; - The global structure of the paper should be reworked for a better readability. If I did well understand, one of the key contributions from (Kasiviswanathan et al. 2011) is the equivalence in term of sample complexity between the Statistical Queries (SQ) formalism of (Kearns 1998) and LDP learning (Lemma 2.3 or Theorem 2.2 & Theorem 2.3 on page 5). As a consequence, LDP Theorem 1.1 is a trivial corollary of SQ Theorem 3.1, and LDP Theorem 1.2 is a consequence of section 4 SQ Lemma's. Wouldn't it simplify the paper to drop (or reduce) the discussions on non-interactive LDP and stick to the essential i.e. the label-non-adaptive SQ learning formalism ? Minor remarks (on short version): line 72: "this is only known" -> "this is the only known" line 76: "algorithms label-non-adaptive" -> "algorithms as label-non-adaptive" line 80: epsilon-LDP should be formalised before line 100: The SQ acronym should be defined here line 248: "Lemma 2.3" -> "Theorems 2.2 and 2.3"

[Author Response · NeurIPS 2019]

**Response to R1:** We thank you for the enthusiastic review and thoughtful comments that we will address in our revision.

**Response to R2:** We thank you for the positive review.

*"Certainly not a breakthrough result however, as separations with slightly stronger assumptions (distribution-specific learning, accessing only local information, separation between SI and Fully Interactive) already existed. "*

We are puzzled by this statement since none of the mentioned works provides any tools or partial progress toward the solution of the problem we consider. Most notably, [JMNR19] considers a separation of completely unrelated nature and is also subsequent to our work. In particular, "slightly stronger assumptions" is not an accurate description of the relationship between these works.

*"Definition 2.1 is actually incorrect."*

Def. 2.1 is the definition of $\epsilon$-LDP from [KLNRS08]. We will clarify that, in the terminology of [JMNR19], it corresponds to compositional $\epsilon$-LDP. The use of this variant of LDP makes our upper bounds stronger and does not affect our lower bounds.

*"The way thm. 1.2 is stated its a bit unclear if its for any class C or a specific class of large-margin linear separators."*

It is stated for any class $C$ and the bound is in terms of the margin complexity of $C$.

*"I wasn't familiar with margin complexity before... so based on 2.5 if the dual class of f contains the all zeros classifier, e.g. x such that $f(x) = 0$ for all X, the margin complexity is infinite?*

In our work Boolean functions are $\{-1, +1\}$-valued (e.g. line 37 or 77) so $C$ cannot include $f(x) = 0$

*"In line 254 shouldn't this be $E[f(x)h(x)] > 1/m$?"*

We believe it's correct as is.

**Response to R3:** We appreciate the reviewer's directness about their lack of familiarity with the area. Our presentation was optimized for readers having basic familiarity with the concept of local differential privacy and interest in this topic.

*"The main (8 pages) version of the paper looks like a hastily truncated version of the 15-pages version provided as supplementary material:"*

That is not true. The 8-page version omits only proofs of some of the results and Section 5 that discusses additional implications of our results.

As usual, our presentation defers the formal definition of the standard concepts to the Preliminaries section (to avoid making the overview even longer). To address some of the reviewer's specific concerns:

1. We will add a pointer to Definition 2.1 (LDP) in the introduction.

2. Add a more explicit definition of the SQ acronym.

[Meta-Review · NeurIPS 2019]

This paper presents the first exponential separation for distribution independent learning of some natural problems in the interactive and non-interactive setting with differential privacy requirements. The reviewers felt this is a very good addition to the work on this topic.